# Dual Circularly Polarized Textile Antenna with Dual Bands and On-/Off-Body Communication Modes for Multifunctional Wearable Devices

Yi Fan [1], Xiongying Liu [2,*], Hongcai Yang [2] and Zhenglin Ju [2]

1   School of Electronics and Information, Guangdong Polytechnic Normal University, Guangzhou 510665, China; fanyi@gpnu.edu.cn
2   School of Electronic and Information Engineering, South China University of Technology, Guangzhou 510640, China
*   Correspondence: liuxy@scut.edu.cn

**Abstract:** A circularly polarized (CP) textile antenna is investigated for concurrent on- and off-body wireless communications in the 2.38 GHz medical body area network and 5.8 GHz industrial, scientific, and medical bands in the wireless body area network. The proposed scheme consists of a square microstrip patch antenna (MPA), in which four shorting pins are employed to tune the two resonate modes of $TM_{10}$ and $TM_{00}$. Notably, the slant corners on MPA are cut symmetrically to realize unidirectional CP radiation, enabling off-body communication. Moreover, four rotating L-shaped parasite elements are loaded to excite the horizontal polarization mode ($TM_{hp}$), which is combined with the $TM_{00}$ mode to implement CP omnidirectional radiation along the human body. For verification, a proof-of-concept prototype with the dimensions of 45 mm × 45 mm × 2 mm was fabricated and characterized. The measured −10 dB impedance bandwidths of 2.5% and 6.7%, the 3 dB AR bandwidths of 2.5% and 2.7%, and the maximum realized gains of −2.8 and 6.8 dBic are achieved in dual bands, respectively. The experimental tests, such as human body loading, structural deformation, and humidity variation, were carried out. In addition, the wireless communication capability was measured and the radiation safety is evaluated. These performances show that the proposed antenna is an appropriate choice for multifunctional wearable applications.

**Keywords:** circular polarization (CP); dual-band antenna; dual modes; wearable applications; wireless body area network

## 1. Introduction

With the speedy development of body area networks (BANs), wearable devices have been becoming necessities in people's daily lives. With the aid of BANs, various wearable portable devices can not only transmit data wirelessly, but also communicate with external data networks, such as mobile communication networks, and wireless sensor networks. This has led to the widespread popularity of wearable devices in many fields, including medical and health [1,2], sports and entertainment [3], and rescue and military [4]. For example, wearable physiological sensors collect important physiological information of patients or elderly people, such as heart rate, blood pressure, glucose concentration, and oxygen concentration, and then transmit it to remote medical devices or mobile terminals for real-time health monitoring, improving people's healthy experience. In terms of sports and entertainment, personal terminal devices with entertainment functions can convey

relevant media data to people's auditory or vision senses, enabling them to experience immersive and personalized entertainment.

The wearable antennas are playing significant roles in realizing wireless communications between BAN nodes [5], wherein the antennas working in multiple bands with radiation pattern diversity are much more desirable for their advantages, such as versatility, space saving, and low cost [6]. Therefore, they are expected to radiate distinct patterns and interconnect different devices mounted on or in proximity of the human body. Compared with the conventional antennas, the design of wearable antennas must overcome several challenges [7], such as guaranteeing their robustness to structural deformation and human body loading, protecting human health within the limitations of specific absorption rate (SAR) levels, wearing conveniently and comfortably, and being immune to polarization mismatch caused by human body movement. Therefore, multiband wearable antennas with several radiation modes, robustness, safety, snugness, and polarization matching are much preferred.

Until now, plenty of dual-band wearable antennas have been proposed. The monopole antennas with a low profile were designed in [8,9]. However, the big SAR may be caused deep in the human tissues due to their backward radiation. To alleviate the coupling between wearable antenna and human body, the patch antenna [10], grounded crossed dipole [11], substrate integrated waveguide antenna [12], and slot antennas [13,14] with full ground plane were presented. In addition, the dual-band wearable antennas loaded with an artificial magnetic conductor (AMC) [15,16] or metasurface [17,18] for a lower SAR were investigated, whereas these dual-band wearable antennas only exhibit a single linear polarization, leading to polarization mismatch when the human body is in motion during data transmission. Thus, the antennas realizing the combination of linearly polarized (LP) and circularly polarized (CP) radiation have been reported in [19]. However, these dual-band wearable antennas are based on mono-mode radiation, only propagating electromagnetic waves on or off the human body. As a result, multifunction wearable applications cannot be implemented.

Hence, dual-band dual-mode wearable antennas have gradually become hotspots due to their capability of simultaneously implementing on-/off-body communications [20–33]. Generally, there are several distinctive technologies for the dual-mode realization of wearable antennas. The first method is achieved by using the stacked patch with a single port [20,21], in which the shorting pins were loaded on the stacked patch to provide radiation pattern diversity. However, they are made of rigid materials. In the second technique, it is a straightforward method to feed two radiating elements separately for the dual-mode radiation [22–25]. The shorted annular ring patch embedded in a circular patch [23] or chamfered rectangular patch [24] was fed through two ports to achieve radiation pattern diversity, making them cumbersome. In [25], dual-mode operation has been implemented with a feeding network, increasing the system complexity. In the third scheme, radiation mode diversity is realized by using the special structure of metallic buttons based on the principle of excitation probe mode and patch mode [26,27], but their installation position is limited. At the same time, the dual-mode antennas with a low profile, one port, and single-element configuration have been investigated in recent years [28–33]. In [28], a half-circular patch with two shorting pins and a ground slot was designed to generate vertical and horizontal monopole-like radiation patterns, but the antenna is rigid. Flexible materials, such as textiles [29–31] and polydimethylsiloxane [32,33], were adopted to design dual-mode wearable antennas. In [29,32], the $TM_{11}$ and $TM_{02}$ modes of circular patch antennas are excited for broadside and conical radiation patterns, respectively. Moreover, radiation mode diversity can be achieved by loading probes [30] or open stubs [31] on the microstrip patch antenna (MPA). However, these antennas suffer from the issue of

a large footprint. In [33], a dual-mode wearable antenna composed of a circular period metasurface and patch was presented for on-/off-body communications, but it inevitably requires a stacked structure.

Although these aforementioned wearable antennas operate in dual bands and have the characteristics of a single mode or dual modes, their polarization immunity is not satisfactory. The reason is that circular polarization cannot be achieved simultaneously in both bands. Recently, a polarization rotation wearable antenna equipped with AMC was first proposed in [34], realizing CP in dual bands. However, the antenna has a high profile and still only achieves single-mode radiation.

Additionally, recent advancements in the integration of flexible devices and intelligent computing technologies have emerged as a research hotspot. In [35], a cognitive-computing-based model is proposed for predicting the flow status of flexible rectifiers, achieving precise fluid control through machine-learning algorithms, with potential applications in the biomedical, chemical, and robotic fields. Furthermore, studies on haptic feedback utilize soft materials and pneumatic actuators to enhance the comfort and tactile feedback capabilities of wearable devices [36]. Hence, future research could leverage machine learning to optimize the design of textile antennas, while exploring the integration of textile antennas with wearable tactile feedback systems.

In this research, a diverse-pattern and CP textile antenna in two bands with a low-profile, single-port, and mono-element configuration is designed for versatile on-/off-body communications. By adopting probes, L-shaped parasite elements, and truncating corners, the omnidirectional CP radiation at 2.38 GHz for realizing the linkage between wearable devices distributed on the different region of the human body and the unidirectional CP radiation at 5.8 GHz for implementing the off-body communication between mobile nodes in BAN are achieved. Moreover, the proposed antenna is cut in a planar textile, making it more convenient for sew-in and panel-based integration with garments. To the best of the authors' knowledge, the dual-band CP textile antenna with dual radiation modes is first proposed.

The remaining contents of this article are organized as follows: Section 2 describes the scheme of the proposed antenna and studies the mechanism of dual radiating modes, miniaturization, and CP actualization. In Section 3, the effect of structural deformation and the variation in substrate properties on antenna performance are thoroughly analyzed, together with the evaluation of wearable safety. Real-situation tests are carried out in Section 4 before the conclusion is given in Section 5.

## 2. Antenna Design and Operating Mechanism

### 2.1. Antenna Configuration

The structure of the proposed dual-band dual-mode CP antenna is shown in Figure 1a–c, which is composed of an MPA, four shorting pins, and four rotating L-shaped parasite elements (RLPEs). The substrate of the antenna uses the flexible felt with a dielectric constant of $\varepsilon_r = 1.2$ and a loss tangent of $\tan \delta = 0.02$. The radiation elements are composed of conductive nylon fabric with a thickness of 0.13 mm and a surface resistivity of less than 0.009 $\Omega$/sq [37].

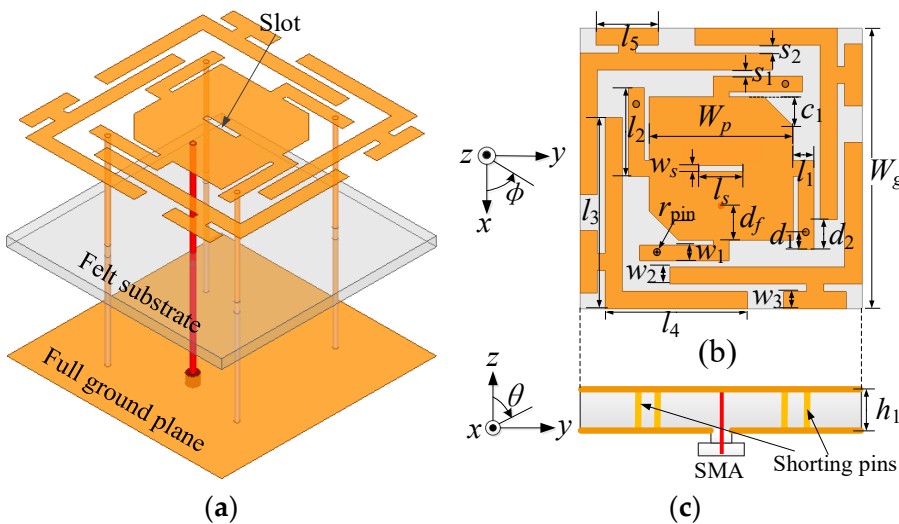

(a)

(b)

(c)

**Figure 1.** Geometry of the developed antenna. (**a**) Perspective view, (**b**) top view, and (**c**) side view. The optimized sizes (mm): $w_g = 45$, $w_p = 23$, $w_s = 1$, $w_1 = 2.5$, $w_2 = 2.8$, $w_3 = 2.8$, $l_s = 7$, $l_1 = 3.2$, $l_2 = 14.2$, $l_3 = 30.5$, $l_4 = 22.8$, $l_5 = 10$, $d_f = 5.5$, $d_1 = 2.5$, $d_2 = 5$, $s_1 = 1$, $s_2 = 1.1$, $c_1 = 4.4$, $h_1 = 2$, and $r_{pin} = 0.5$.

In order to excite the $TM_{00}$ and $TM_{10}$ modes of the MPA and realize miniaturization, four shorting pins are extended to the four sides of MPA by stubs. Through cutting the slant corners on the MPA symmetrically, a pair of degenerated modes (i.e., $TM_{10}$ and $TM_{01}$) are generated in the upper band, achieving unidirectional CP radiation. Moreover, four RLPEs are loaded to excite the horizontal polarization mode ($TM_{hp}$) in the lower band, which is combined with the $TM_{00}$ mode to implement omnidirectional CP radiation. Notably, a slot is cut in the middle of the MPA to neutralize the equivalent inductance caused by the shorting pins and improve the impedance matching. With the assistance of ANSYS HFSS v.21, the optimized values of the geometrical parameters are given in the caption of Figure 1.

Considering that the suggested antenna is worn on the human body, the lossy tissues should be regarded as the loading in the design. Hence, a three-layer phantom is built, as demonstrated in Figure 2, comprising skin, fat, and muscle. The dielectric properties and dimensions of each tissue are listed in Table 1 [38]. To imitate wearing scenarios, the textile material with a thickness of $d_0$ (5 mm) is filled between the proposed antenna and the digital phantom.

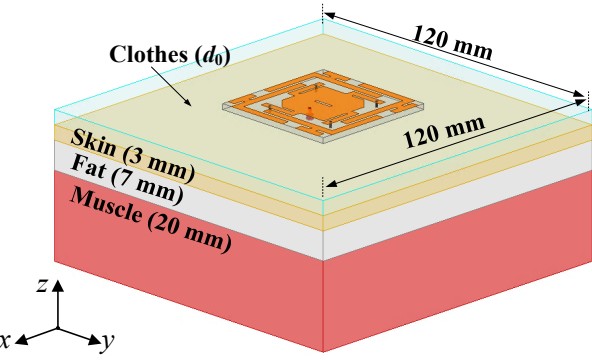

**Figure 2.** Three-layer homogeneous phantom in the simulation.

**Table 1.** Dielectric characteristics and sizes of each tissue.

| Tissue | 2.38 GHz | | 5.8 GHz | | Thickness (mm) |
|---|---|---|---|---|---|
| | $\varepsilon_r$ | $\sigma$ (S/m) | $\varepsilon_r$ | $\sigma$ (S/m) | |
| Skin | 38.09 | 1.43 | 35.11 | 3.717 | 3 |
| Fat | 5.29 | 0.10 | 4.95 | 0.29 | 7 |
| Muscle | 52.82 | 1.69 | 48.48 | 4.96 | 20 |

*2.2. Implementation of Dual Bands and Miniaturization*

As analyzed in a previous work [30], the dual-band characteristic is achieved by introducing shorting pins on one radiation patch, in which the $TM_{00}$ and $TM_{10}$ resonant modes of the MPA can be tuned by changing the spacing and radius of shorting pins. However, it radiates single-CP electromagnetic waves and has a large footprint. To minimize the antenna size, the proposed structure is evolved from Case 1 to Case 3, as demonstrated in Figure 3a–c. The corresponding input impedance variation and schematic electric field distributions [39] are introduced to characterize the resonant mode of the antenna, as displayed in Figure 4a,b. First, four shorting pins are loaded on the edge of the MPA, and two resonate modes are separately generated near 4.9 and 6.8 GHz in Case 1. Second, as four shorting pins are extended to the four sides of MPA, the resonant frequencies of two modes shift down due to the lengthening of the current path. However, the antenna structure in Case 2 occupies a large area. Third, four stubs in Case 3 are bent to realize the required dual-band operation while obtaining the compact structure.

According to the electric field distribution in Figure 4b, it can be seen that the radiation mode of $TM_{00}$ is similar to that of a monopole, with a quarter wavelength resonance. The $TM_{10}$ mode is the main mode of the microstrip patch antenna, which is a half-wavelength resonance. Therefore, the resonant $TM_{00}$ and $TM_{10}$ mode can been expressed as follows:

$$f_{00} = \frac{c}{4L_{e1}\sqrt{\varepsilon_e}}, \tag{1}$$

$$f_{10} = \frac{c}{2L_{e2}\sqrt{\varepsilon_e}}, \tag{2}$$

where $c$ is the speed of light in free space, $L_{e1}$ and $L_{e2}$ are the effective electrical lengths of $TM_{00}$ and $TM_{10}$ modes, respectively, and $\varepsilon_e$ is the effective dielectric constant obtained from (3):

$$\varepsilon_e = \frac{\varepsilon_r + 1}{2} + \frac{\varepsilon_r - 1}{2}\left(1 + \frac{12h_1}{w_p}\right)^{-1/2}. \tag{3}$$

Moreover, by means of the current distribution diagram in Figure 5a,b, the effective electrical lengths $L_{e1}$ and $L_{e2}$ in (1) and (2) can be approximately written as follows:

$$L_{e1} = \frac{w_p}{2} + l_1 + l_2 - d_1 + l_{pin}, \tag{4}$$

$$L_{e2} = w_p + 2\Delta\xi, \tag{5}$$

$$l_{pin} = \alpha \ln\left(\frac{\beta}{2\pi r_{pin}}\right), \tag{6}$$

$$\Delta\xi = \gamma(l_1 + l_2 - d_1 + l_{pin}), \tag{7}$$

among the four formulae above, $w_p$, $l_1$, $l_2$, $d_1$, and $r_{pin}$ are the antenna structural parameters (as shown in Figure 1), $l_{pin}$ is the equivalent electrical length induced by the shorting pins [40], $\Delta\xi$ is the equivalent extension length on four sides of the MPA, and $\alpha$, $\beta$, and

$\gamma$ are the correction factors, whose values can be determined to be 3.95, 3.2, and 0.085, respectively, with the aid of numerical analysis methods. Therefore, by using the formulae from (1) to (7), the dimensions of the miniaturized dual-band antenna can be quickly and approximately calculated.

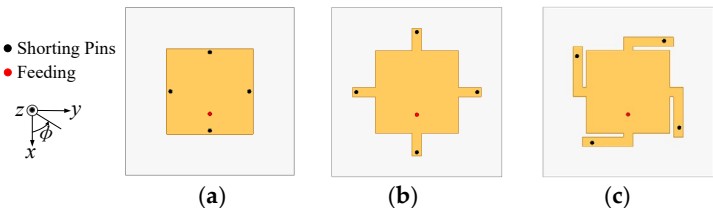

**Figure 3.** Evolution of antenna miniaturization from (**a**) Case 1, (**b**) Case 2, to (**c**) Case 3.

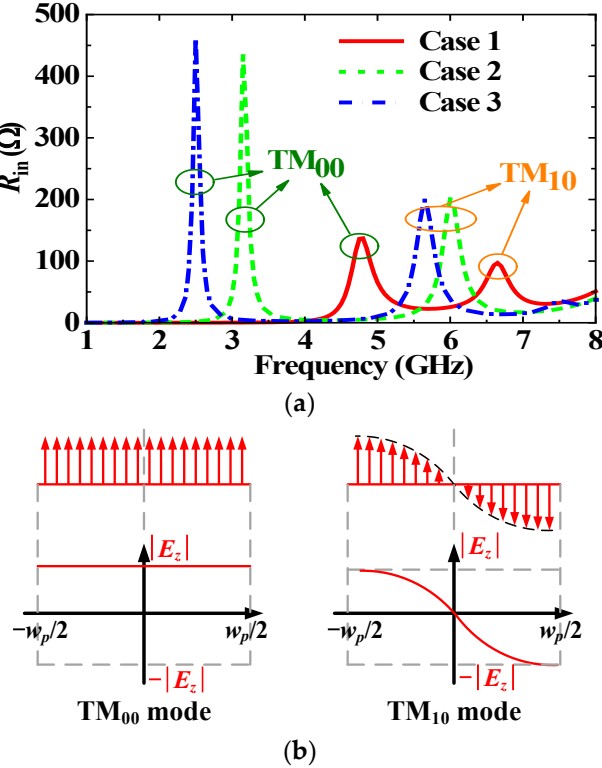

**Figure 4.** (**a**) Input resistance $R_{in}$ of $TM_{00}$ and $TM_{10}$ modes in each case, and (**b**) schematic electric field distributions in Case 1.

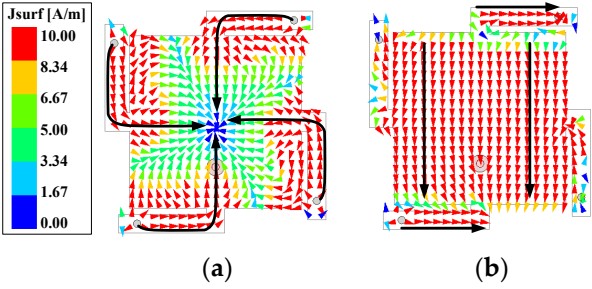

**Figure 5.** Current distributions in Case 3: (**a**) $TM_{00}$ mode @2.38 GHz, and (**b**) $TM_{10}$ mode @5.8 GHz.

### 2.3. Analysis of Dual-Mode CP Actualization

Theoretically, the implementation of CP radiation needs two orthogonal modes to be excited at the same time. In the upper band, by symmetrically cutting off slant corners on the MPA (i.e., Case 3), the $TM_{10}$ mode is separated into Modes 1 and 2, as demonstrated

in Figure 6a. It can be seen from Figure 6b,c that Mode 1 is a quasi-$TM_{10}$ mode and Mode 2 is an approximate $TM_{01}$ mode which are a pair of degenerate modes. Additionally, the electric field distribution in Figure 6d proves that Mode 3 in the lower band is a vertical monopole-like mode of $TM_{00}$. Therefore, it is inevitable that we generate a horizontally polarized ring current mode of $TM_{hp}$ to realize omnidirectional CP radiation in the lower band [41]. In this design, four RLPEs are loaded around the MPA, as shown in Figure 7a, generating extra modes of Modes 4 and 5. With reference to the current distributions in Figure 7b,c, it can be concluded that Mode 4 is the required mode of $TM_{hp}$, while Mode 5 is the high-order mode that will be suppressed.

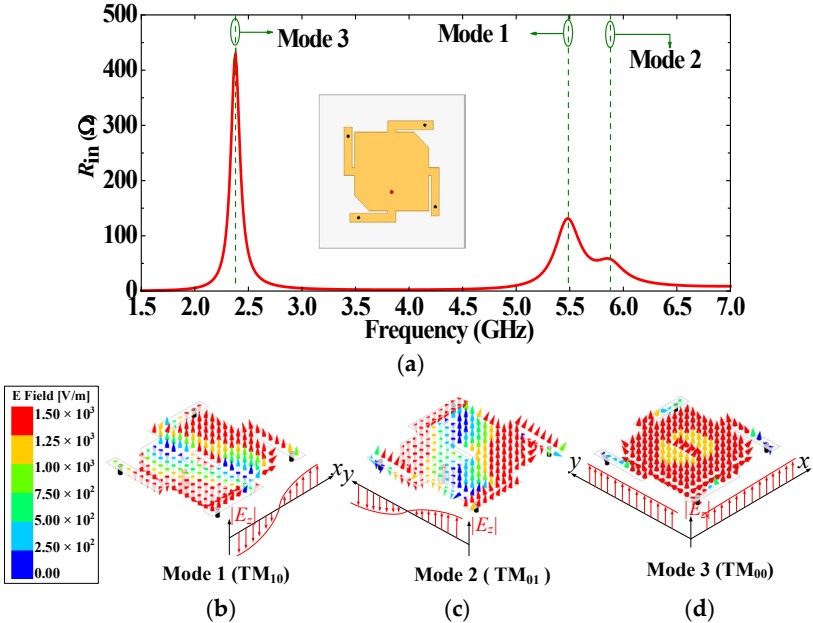

**Figure 6.** (**a**) Input resistance of symmetrically cutting off slant corners in Case 3; and electric field distributions of (**b**) Mode 1, (**c**) Mode 2, and (**d**) Mode 3.

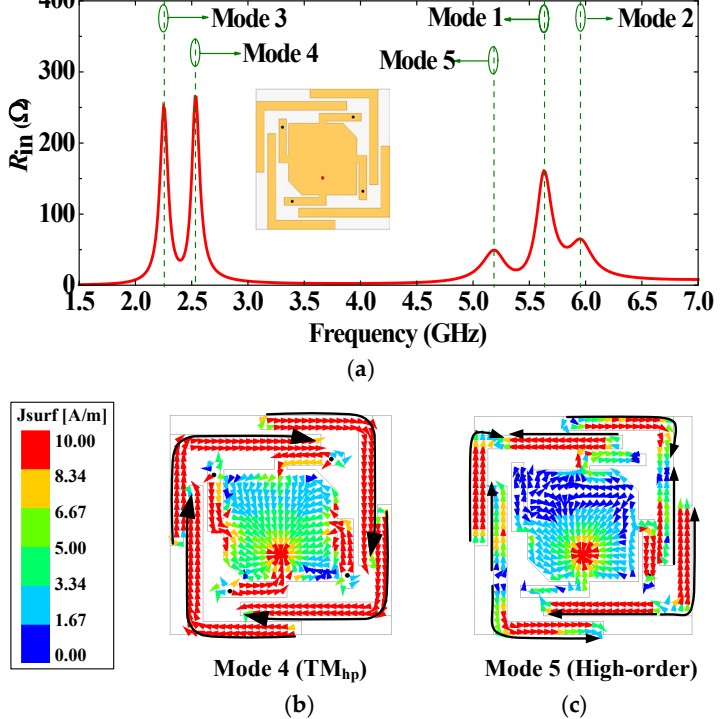

**Figure 7.** (**a**) Input resistance, and current distributions of (**b**) Mode 4 and (**c**) Mode 5.

*2.4. Analysis of Mode 5 Suppression*

Based on the above mode analysis, Mode 5 of original antenna Ant 1 in Figure 8a is near the upper band, which seriously affects the gain of the upper band, as shown in the red solid curve in Figure 9a,b. Figure 9b employs dual vertical axes with distinct meanings to present two types of data. For clarity, directional arrows are added to each curve: Curves with left-pointing arrows indicate variations in AR, while curves with right-pointing arrows demonstrate changes in Gain. For figures with directional arrows in subsequent sections, the information reflected by each curve can also be interpreted based on the arrow orientation and the meaning of the vertical axes. To address the issue, Mode 5 is tuned by loading shorting pins or T-shape stubs on the RLPEs, as displayed in Figure 8b,c. Notably, the shorting pins or T-shape stubs are loaded on the position with the maximum electric field in Mode 5. In Ant 2, the loading of the shorting pins shifts the resonant frequency of Mode 5 to near 2.9 GHz due to the extension of the current path, eliminating the radiation null in the upper band, as exhibited in the green dashed curves in Figure 9a,b, whereas the AR of Ant 2 in the lower band is deteriorated, and Mode 5 still causes interference to the 3 GHz band. In Ant 3, Mode 5 is shifted to near 4.65 GHz by loading T-shape stubs, achieving stable gain in the upper band while suppressing Mode 5 resonance, as shown in the blue dot-dash curves in Figure 9a,b.

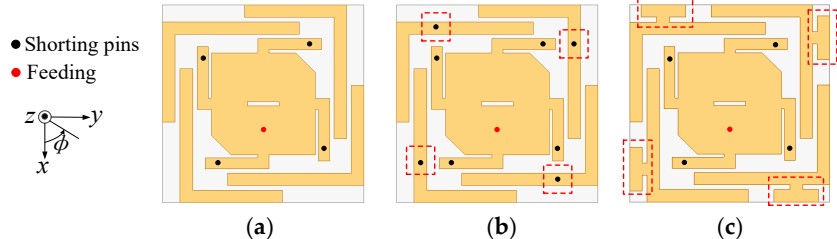

**Figure 8.** Configuration of the proposed antenna evolving from (**a**) Ant 1, (**b**) Ant 2 (shorting pin loading), to (**c**) Ant 3 (T-shape stub loading).

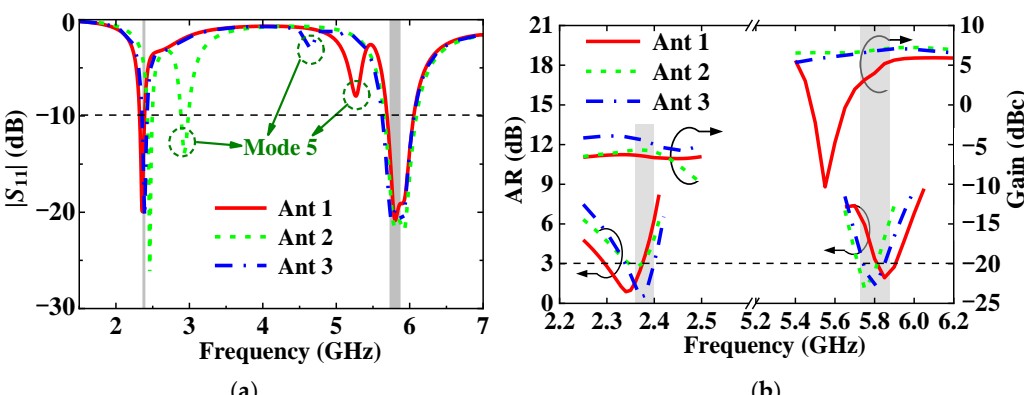

**Figure 9.** Simulated results of three different antennas on phantom, (**a**) $|S_{11}|$, and (**b**) AR and gain.

In summary of the above content, the design mechanism of the dual-band circularly polarized textile antenna with dual modes can be acquired. First, starting from two resonant frequencies, a set of approximate formulae are adopted to obtain a dual-band square-shaped MPA with four bent stubs and four shorting pins. Second, the MPA is symmetrically chamfered to achieve CP unidirectional radiation in the upper band. Third, four RLPEs with a T-stub are loaded around the MPA to achieve omnidirectional CP radiation in the lower band. Finally, in order to adjust impedance matching, a slot is etched in the middle of the MPA.

## 3. Antenna Performance Evaluation

### 3.1. Simulated Results on Phantom

To demonstrate the excellent performance, the proposed antenna is simulated on the phantom of Figure 2. The simulated normalized radiation patterns of the proposed antenna at 2.38 and 5.8 GHz are plotted in Figure 10a,b. It is found that omnidirectional CP radiation at 2.38 GHz and unidirectional CP radiation at 5.8 GHz are achieved, which are desirable for simultaneous on-/off-body communications, respectively. Notably, a cross-polarization ratio of greater than 15 dB is achieved in the dual bands, implying good polarization purity. In addition, the AR and gain beamwidths in the dual bands are presented in Figure 10c,d. It is observed from Figure 10c that the ripple of omnidirectional radiation at 2.38 GHz is less than 1.9 dB, and the 3 dB AR beamwidth covers the whole azimuth plane. Moreover, with reference to Figure 10d, the half-power beamwidth (64°) can be completely covered by the 3 dB AR beamwidth (78°), indicating that CP performance can be guaranteed in the whole effective radiation range.

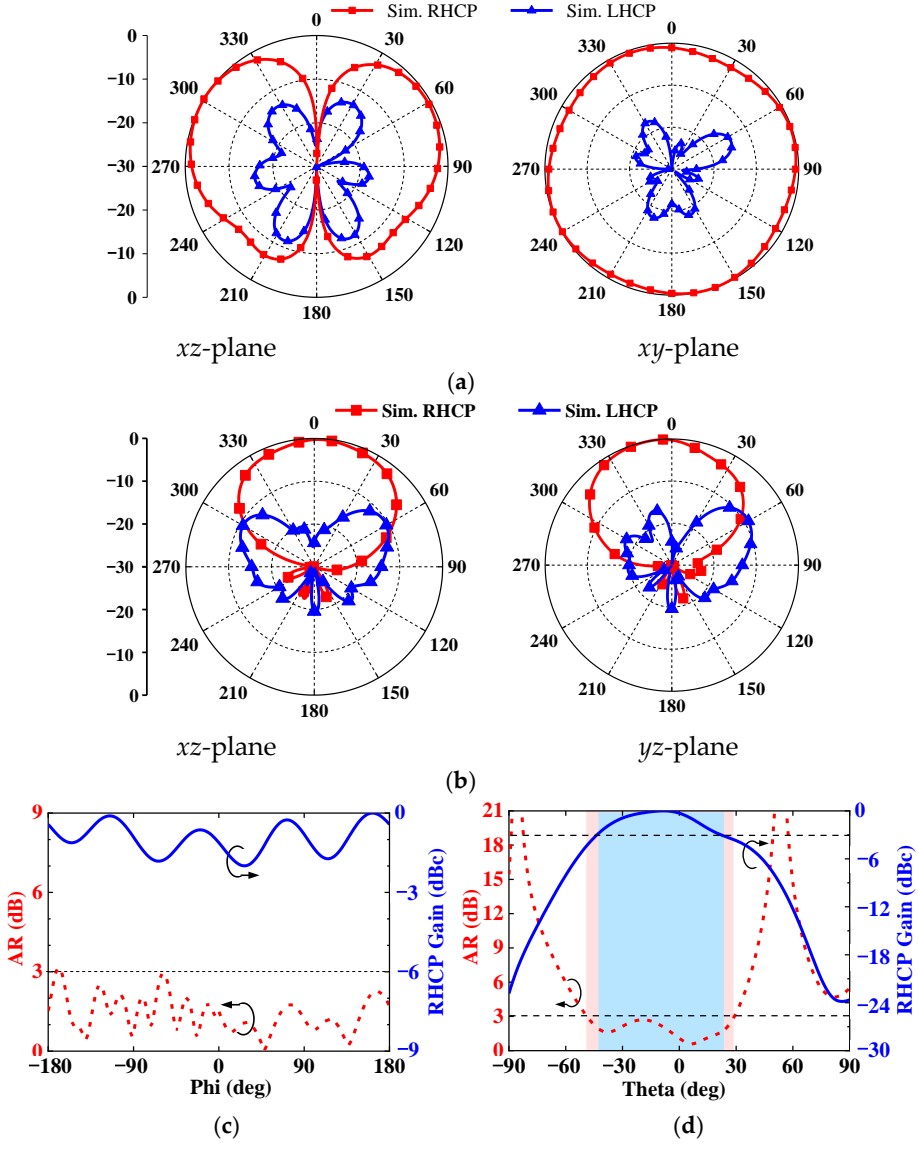

**Figure 10.** Simulated results of the suggested antenna on the phantom, and normalized radiation patterns at (**a**) 2.38 GHz and (**b**) 5.8 GHz; and AR and gain beamwidths at (**c**) 2.38 GHz and (**d**) 5.8 GHz.

### 3.2. Robustness to Structural Deformation

Due to the fact that the outline of the human body is cambered, and even diverse regions have different curvatures, the wearable antennas should be conformal to the body for comfort. For the convenience analysis, the proposed antenna is deformed along the cylindrical phantom with a curving radius, as shown in Figure 11a. Moreover, Figure 11b,c present a sketch map of the antenna bending along the *x*- and *y*-axes, wherein three typical curvatures, i.e., $R_{x/y}$ = 50, 75, and 100 mm, are selected, corresponding to the installing positions on the upper arm, leg, and chest, respectively. With reference to Figure 12a–d, it can be concluded that the impedance match of the antenna is slightly changed, but the gain in the lower band is fluctuant as the antenna is bent along the *x*- or *y*-axes. This is due to the fact that the cylindrical human phantom has different reflection effects on the omnidirectional radiation pattern, resulting in a gain ripple. Additionally, it can be seen from Figure 12d that the AR shifts up in the lower band as the antenna is curved along the *y*-axis due to the influence of bending along the *y*-axis on the current paths of Modes 3 and 4, leading to a small shift in the AR.

In general, the human body exhibits dynamic changes during movement, which may cause structural deformations in the antenna and thereby affect its performance. This work studies the deformation of the antenna along the *x*-axis and *y*-axis, which can accurately reflect the impact of dynamic motion on antenna performance. As demonstrated by the simulated results in Figure 12a–d, the proposed antenna retains its circular polarization characteristics without degradation, and all aspects of its performance remain stable during body movements.

In practical wearable scenarios, the distance $d_0$ between the proposed antenna and the human body may vary due to body movements. Therefore, the effects of $d_0$ variations on antenna performance are investigated, as shown in Figure 12e,f. Owing to the effective isolation of electromagnetic coupling between the antenna and the human body by the complete ground plane, the antenna maintains excellent impedance matching and circular polarization performance even under significant fluctuations in $d_0$.

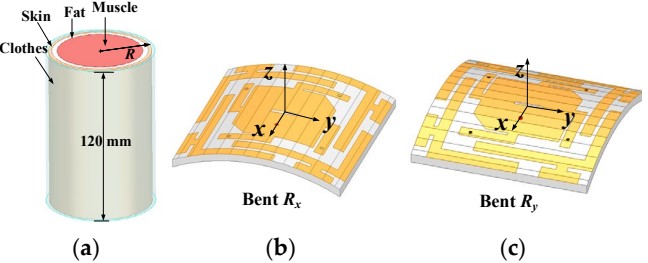

(a)  (b)  (c)

**Figure 11.** (**a**) Schematic of the cylindrical phantom; and the proposed antenna bending along (**b**) the *x*-axis, and (**c**) the *y*-axis.

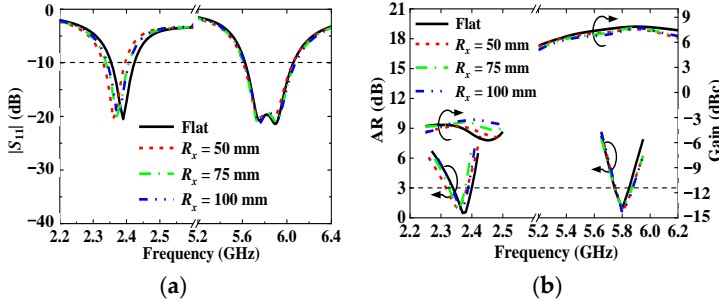

(a)  (b)

**Figure 12.** *Cont.*

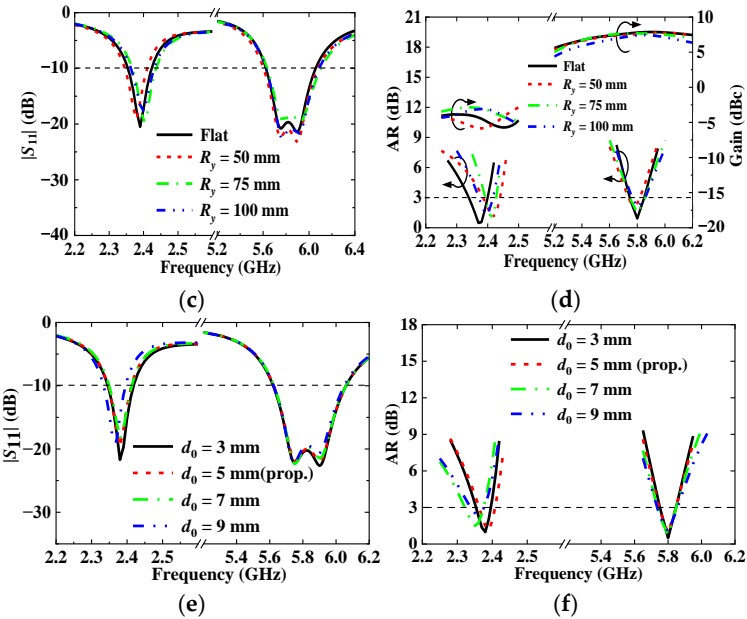

**Figure 12.** (**a**) $|S_{11}|$, and (**b**) AR and gain when bending along the *x*-axis; (**c**) $|S_{11}|$, and (**d**) AR and gain when bending along the *y*-axis; and (**e**) $|S_{11}|$, and (**f**) AR when varying with distance $d_0$ between the antenna and human body.

### 3.3. Influence of Substrate Properties

In real wearing scenarios, the performance of the proposed antenna may be affected by the reduction in substrate thickness due to environment pressure and by the variation in dielectric properties caused by environmental humidity. To analyze the robustness of the proposed antenna, the effects of a ±5% difference in relative permittivity, 10% variation in the thickness of the substrate, and ±10% change in textile conductivity on the performance of the proposed antenna are studied. With reference to Figure 13a,b, the frequency shift occurs in the pass band of the impedance match, AR. It can be seen that, as the relative permittivity increases, the resonant frequencies decrease, but the proposed antenna can essentially maintain good impedance matching and high CP purity in the dual bands. In addition, as shown in Figure 13c,d, the decrease in substrate thickness causes a slight frequency shift in the lower band but without affecting the radiation gain, reflecting that the proposed antenna is tolerant to the variations in substrate properties caused by uncontrollable factors. With reference to Figure 13e,f, the impedance matching, AR, and gain of the antenna are not affected by changes in conductivity. Therefore, the proposed antenna can essentially maintain good impedance matching and high CP purity in the dual bands.

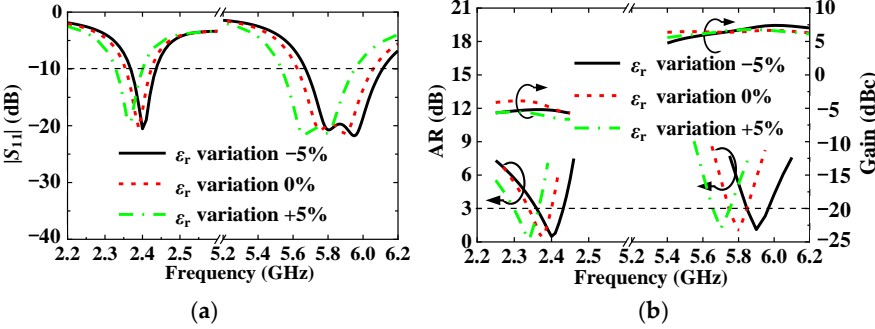

**Figure 13.** *Cont.*

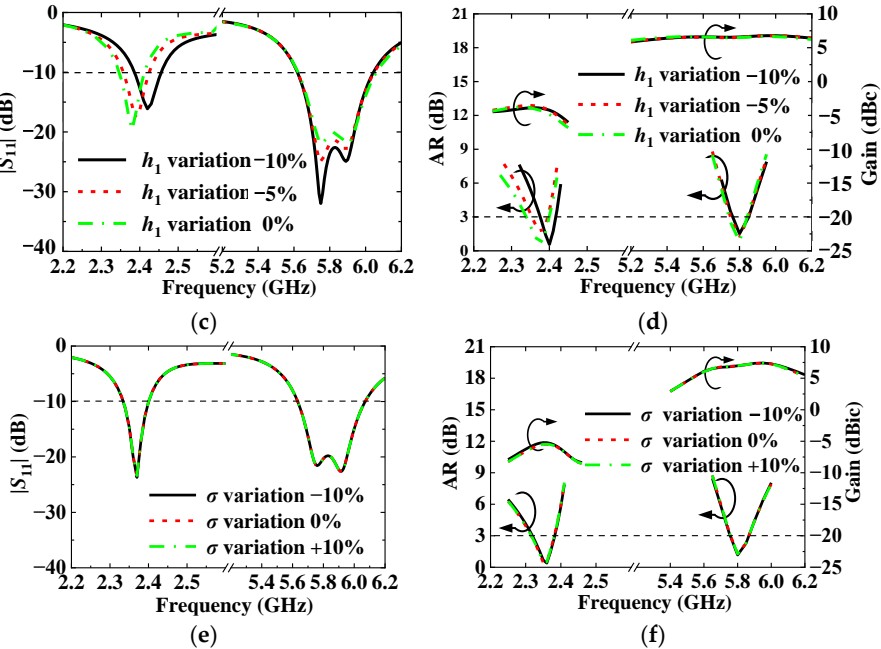

**Figure 13.** (**a**) $|S_{11}|$, and (**b**) AR and gain when varying with relative permittivity of the substrate material; (**c**) $|S_{11}|$, and (**d**) AR and gain when varying with thickness of the substrate material; and (**e**) $|S_{11}|$, and (**f**) AR and gain when varying with conductivity of the textile.

### 3.4. Evaluation of Radiation Safety

Due to the fact that the wearable antenna is worn on the human body, the electromagnetic waves would inevitably leak into the human body, endangering the health of the human body. Thus, the SAR should be assessed to ensure that the radiation of the proposed antenna is safe. Herein, a Hugo voxel model is integrated into the CST Microwave Suite, imitating real scenarios. The gap between the voxel model and antenna is set to be about 5 mm, which is similar to the foregoing setup. As a benchmark, the SAR is calculated according to (8) by transmitting an incident power of 100 mW [42]

$$SAR = \frac{\sigma |E|^2}{\rho} \tag{8}$$

where $E$ represents the electric field strength in the tissue (V/m), $\sigma$ denotes the conductivity of the tissue (S/m), and $\rho$ is the mass density of the tissue (kg/m$^3$).

As shown in Figures 14 and 15, the 1 g tissue averaged SARs of the designed antenna at 2.38 and 5.8 GHz are studied, in which the proposed antenna is worn on different positions of the human body, including the upper arm, leg, and chest. It can be found that, in all three scenarios, the averaged SARs in 1 g tissue of the dual bands range from 0.343 to 0.592 W/kg with a peak value of 0.592 W/kg and from 0.215 to 0.287 W/kg with a maximum value of 0.287 W/kg, respectively. In other words, if the maximum allowable input power is not over 270 mW (24.3 dBm), the SAR can be kept within a safe range (i.e., below 1.6 W/kg).

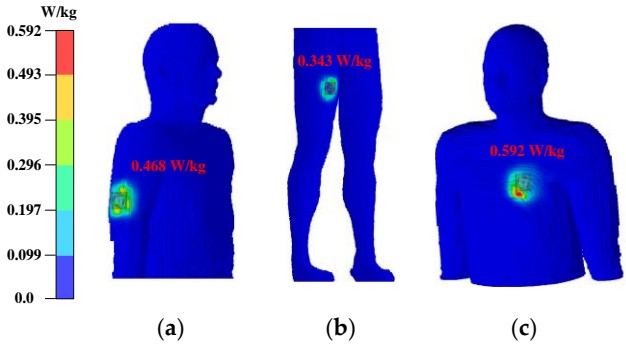

**Figure 14.** Simulated 1 g tissue averaged SAR distributions of the proposed antenna at 2.38 GHz on (**a**) upper arm, (**b**) leg, and (**c**) chest.

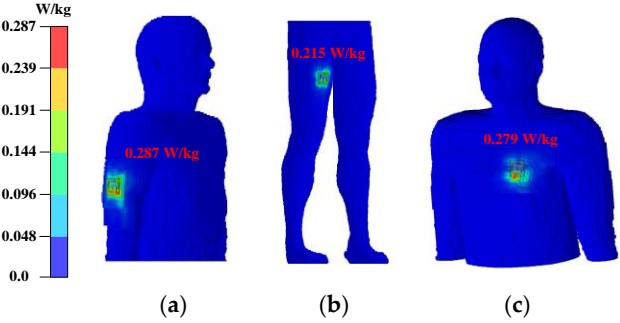

**Figure 15.** Simulated 1 g tissue averaged SAR distributions of the proposed antenna at 5.8 GHz on (**a**) upper arm, (**b**) leg, and (**c**) chest.

## 4. Experimental Results and Discussion

As a proof of concept, the prototype of the proposed textile antenna was fabricated (see Figure 16a–c), and conductive nylon fabric and felt substrate, both widely utilized in antenna production, serve as common materials. The manufacturing process comprises four steps: First, the conductive nylon fabric via laser ablation is cut to fabricate the radiation element and ground plane; second, these components are glued to the top and bottom of the felt material, which have been featured with four shorting holes and one hole for feeding; third, four wires are threaded through shorting holes, establishing a connection between the top and bottom layers of felt; and, lastly, an SMA connector is soldered onto the combined layers, enabling port excitation. Two male volunteers (1.73 and 1.78 m in height, 62 and 70 kg in weight) were involved when the measurement setups were built. The reflection and transmission coefficients were measured on a piece of fresh pork and human body through the Agilent N5230A vector network analyzer (VNA), respectively. The radiation characteristics in the far field were tested on a piece of fresh pork in the anechoic chamber.

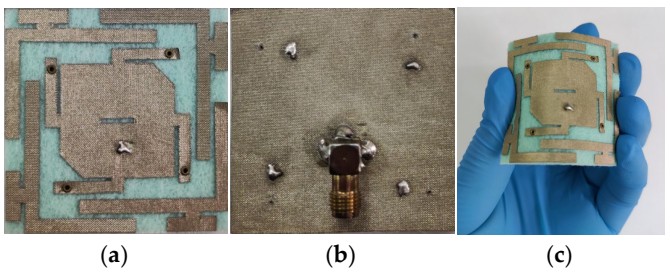

**Figure 16.** Photographs of a fabricated textile antenna in (**a**) top view, (**b**) bottom view, and (**c**) bending state.

### 4.1. Real Performance Test

The computed and measured $|S_{11}|$ of the proposed antenna on a flat phantom is compared in Figure 17a. Herein, different textile materials, such as felt, cordura, polyester, and jeans, were filled between the proposed antenna and a piece of fresh pork, evaluating the impacts of clothes. The measured results indicate that the effect of different garment fillings on antenna performance is slight. On average, the measured −10 dB impedance bandwidths are 2.5% (2.34–2.40 GHz) in the lower band and 6.7% (5.59–5.98 GHz) in the upper band, respectively. Notably, compared with the simulated results, the measured ones drifted down by about 20 MHz due to the offset introduced by the fabrication errors and soldering parts.

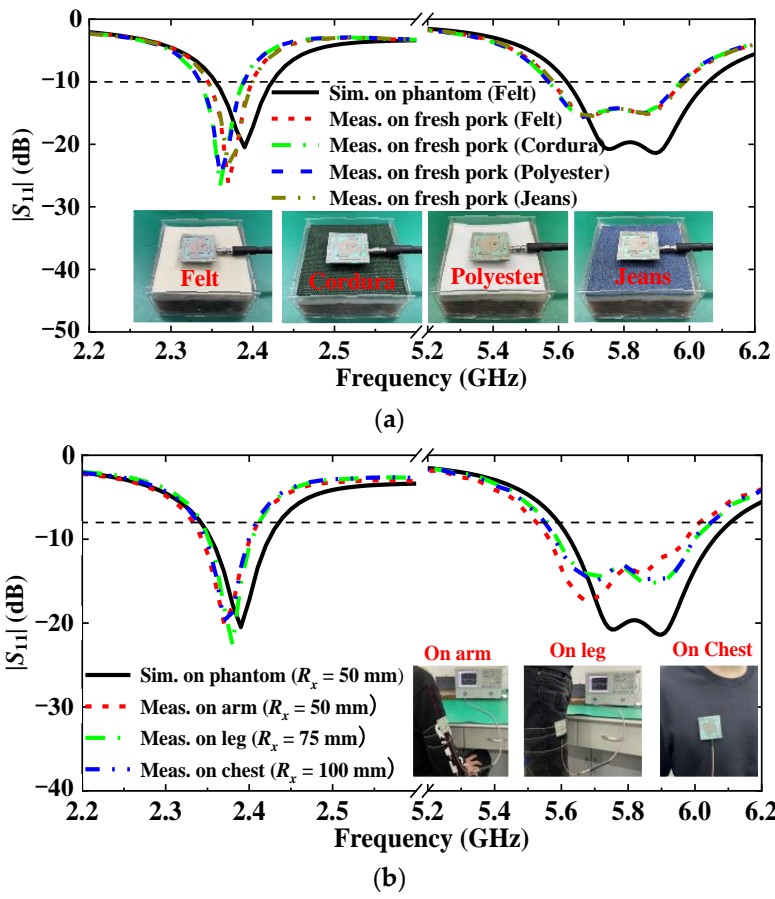

**Figure 17.** Simulated and measured $|S_{11}|$ of the proposed antenna with (**a**) different textile materials loading and (**b**) structural deformation.

As the human body is a curved object, the antenna cannot keep a flat shape when being worn. Hence, the $|S_{11}|$ of the proposed antenna was measured on the different regions of the human body, including the upper arm, leg, and chest, corresponding to $R_x$ = 50, 75, and 100 mm, respectively, as depicted in Figure 17b. It is found that the measured impedance performance remains stable in different bending scenarios and fully covers the required dual bands, which implies that the proposed antenna is robust against curvature and can be worn on different regions of the human body.

Moreover, considering that textiles absorb water easily in a wet environment, the study on the influence of different humidity conditions on textile antenna performance was carried out. With reference to the inset of Figure 18, the experimental setup was established, where the humidifier was employed to change the relative humidity (RH) in a sealed box. First, with the background of 45% RH at room temperature, the RH is increased by 10−20% at each step and preserved for 10 min. Then, the $|S_{11}|$ was measured and recorded by the

VNA. As observed in Figure 18, with the rise in RH, the resonant frequency of the antenna drifts down slightly, exhibiting that the proposed antenna has a tolerance to low or high humidity caused by environmental change.

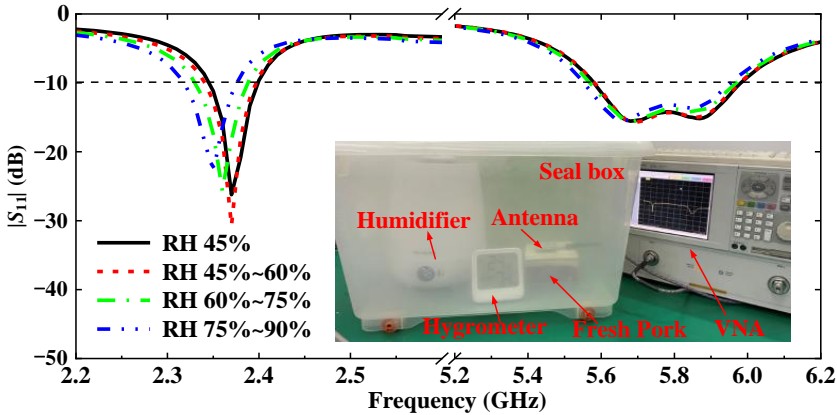

**Figure 18.** Measured $|S_{11}|$ in various humidity environment.

*4.2. Communication Capability Measurement*

To study the wireless communication ability on the body in the lower band, two identical antennas were mounted on different parts of the human body. As shown in Figure 19a–c, an antenna was placed on the chest and the other was bound on the wrist, arm, and leg, respectively. It can be seen from Figure 19d that the on-body propagation loss changes slightly in the lower band since the path loss in the lower band is smaller than that in the upper band. Hence, the proposed antenna provides a more reliable link in the lower band and the transmission losses are around 42–48 dB, which is suitable for on-body communications.

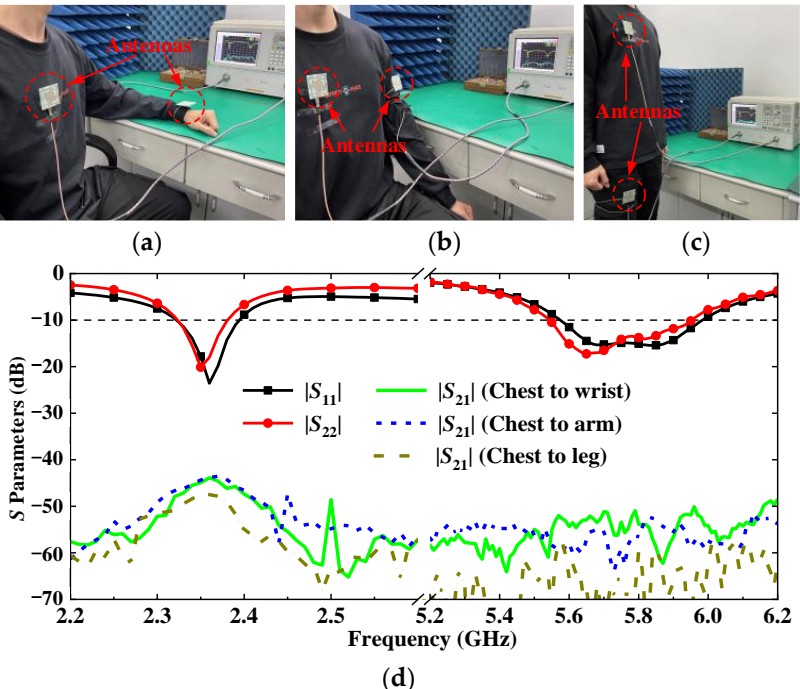

**Figure 19.** On-body transmission measurement setups and measured results: (**a**) chest to wrist, (**b**) chest to arm, (**c**) chest to leg, and (**d**) *S*-parameters.

To evaluate the off-body communication in the upper band, the proposed two antennas were mounted on the chests of two volunteers, as shown in the insets of Figure 20. Without

losing generality, the distances between the antennas are set to 1 m, and the transmission losses were tested in face-to-face and face-to-side scenarios. With reference to Figure 20, the transmission loss of about 30 dB in the face-to-face scenario is 16 dB lower than that in the face-to-side situation. With reference to Figure 10b, this is mainly due to the increased path loss from the face–face link to the face–side link.

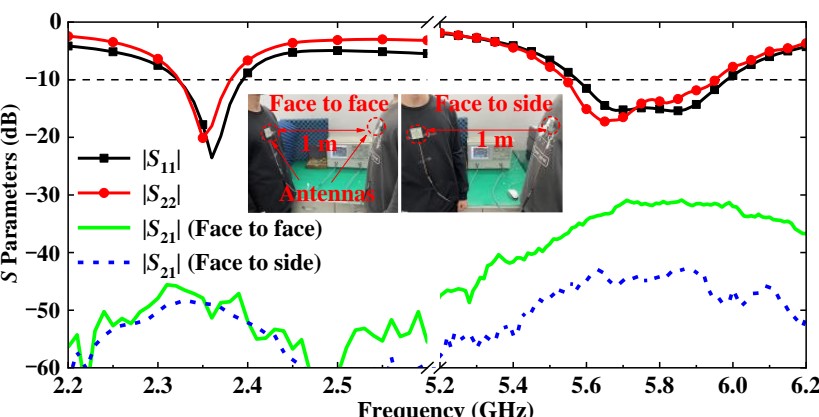

**Figure 20.** Measured *S*-parameters in different off-body communication scenarios.

Based on the SAR evaluation in Section 3.4, the maximum input power of the antenna can reach 24.3 dBm. Assuming that the minimum receiver sensitivity is −75 dBm for 5.8 GHz ISM band applications 26, then the maximum link loss of 99.3 dB is acceptable. Therefore, the off-body link with a loss of less than 50 dB is quite reliable in the upper band even if the proposed antenna is used in the face-to-side situation.

### 4.3. Radiation Characteristics Verification

With reference to Figure 21a–d, the simulated and measured ARs, gains, efficiencies, and radiation patterns of the proposed antenna on the phantom are compared. As illustrated in Figure 21a, a considerably low AR is achieved in the dual bands, and the measured 3 dB AR bandwidths of the two bands are 2.5% (2.317–2.377 GHz) and 2.7% (5.658–5.814 GHz), respectively. It is noted that the frequency of the measured results shifts down compared with the simulated ones, which is consistent with the measurement errors of the reflection coefficient. In addition, as shown in Figure 21b, the measured gain and efficiency of the proposed antenna are consistent with the simulated ones, and the slight divergence between the results are mainly due to the fabrication tolerances and the difference between the numerical phantom and fresh pork. The measured peak realized gains in the two bands are −2.8 and 6.8 dBic with the radiation efficiencies of 25.2% and 67.7%, respectively. The divergence between the simulated and measured results can be attributed to the fabricating tolerance and the phantom inaccuracy.

To evaluate the radiating properties, the simulated normalized radiation patterns and the measured ones of the antenna are analyzed. It can be observed from Figure 21c that the RHCP monopole-like radiation patterns with a ripple of less than 5.4 dB are obtained at 2.38 GHz, which is desirable for on-body communications. Moreover, as shown in Figure 21d, the RHCP unidirectional radiation patterns are fulfilled at 5.8 GHz with a cross-polarization ratio of greater than 20 dB in the broadside direction for off-body communications.

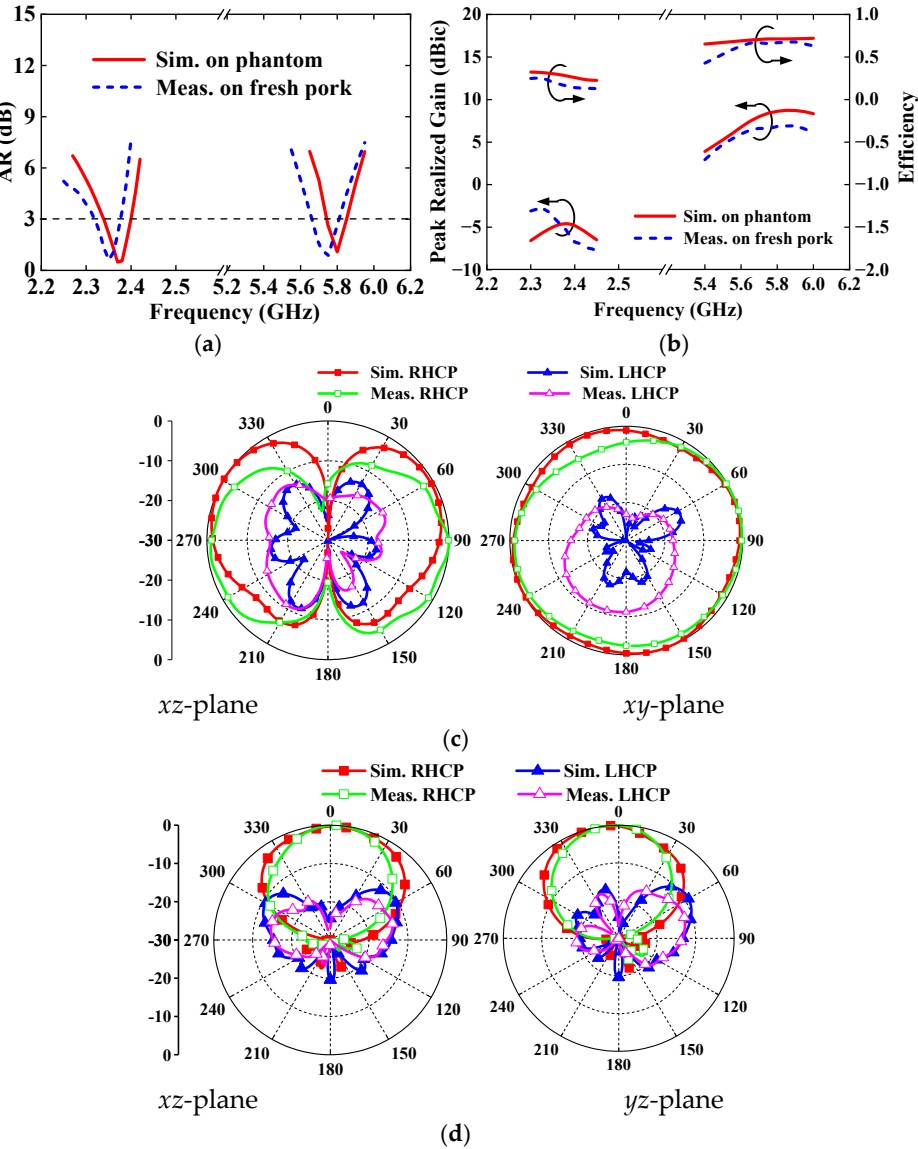

**Figure 21.** Simulated and measured results of the proposed antenna on phantom: (**a**) ARs, (**b**) peak realized gains and efficiencies, (**c**) normalized radiation patterns at 2.38 GHz, and (**d**) normalized radiation patterns at 5.8 GHz.

Although the previously reported similar works in [8,19–21], and [24] also have negative gains as listed in Table 2, the performance of the proposed antenna is studied in free space to clarify why the proposed antenna has low gain. As shown in Figure 22, the proposed antenna in free space has a positive peak realized gain of 0.69 dBic and a high radiation efficiency of 88% at the lower central frequency of 2.38 GHz, whereas the simulated negative gain of −4.59 dBic and radiation efficiency of 28% are obtained as the proposed antenna is worn on the phantom. This can be mainly attributed to the following reasons: (1) the proposed antenna working in the lower band has smaller electrical sizes of $0.36\lambda_0 \times 0.36\lambda_0 \times 0.016\lambda_0$ ($\lambda_0$ is the free space wavelength at 2.38 GHz) than its operating in the upper band; (2) due to the large radiation range, the antenna with the omnidirectional radiation pattern in the lower band would have a lower directivity; and (3) the omnidirectional antenna radiates energy into the lossy human body, causing electromagnetic energy to be absorbed.

**Table 2.** Comparison with previously reported wearable antennas.

| Ref. | $f_1$ & $f_2$ (GHz) | Dimensions ($\lambda_0 \times \lambda_0 \times \lambda_0$) | Bandwidth $|S_{11}|<$ $-10$ dB | Polarization | Radiation Pattern | Peak Realized Gain (dBic) | Number of Layers and Ports | Type of Antenna | Material ($\varepsilon_r$, tan $\delta$) |
|---|---|---|---|---|---|---|---|---|---|
| [8] | 2.45/5.8 | $0.47 \times 0.33 \times 0.001(0.75)$ * | 60%/31.4% | LP/LP | O/O | $-2.3/1.22$ | 1 layer & 1 port | Flexible | Kapton (3.5,0.002) |
| [15] | 3.5/5.8 | $1.0 \times 1.0 \times 0.053(25.6)$ * | 29.1%/8.7% | LP/LP | U/U | 9.3/6.6 | 2 layers & 1 port | Semi-flexible | Felt (1.22,0.016) |
| [19] | 2.45/5.8 | $\pi \times 0.2 \times 0.2 \times 0.04(2.4)$ * | 10%/9% | LP/CP | O/O | $-5.1/3.3$ | 2 layers & 1 port | Flexible | Felt (1.63,0.044) |
| [20] | 2.38/5.8 | $0.18 \times 0.18 \times 0.04(0.6)$ * | 2.8%/9% | LP/CP | O/U | $-4.5/3.8$ | 2 layers & 1 port | Rigid | FR4 (4.4,0.02) |
| [21] | 2.45/3.65 | $0.36 \times 0.25 \times 0.026(1.1)$ * | 4.9%/2.8% | LP/LP | U/O | $0.6/-1.6$ | 2 layers & 1 port | Rigid | FR4 (4.4,0.02) |
| [23] | 2.45/5.8 | $\pi \times 0.28 \times 0.28 \times 0.013(1.5)$ * | 3.5%/3.4% | LP/LP | O/U | 0.8/5.4 | 1 layer & 2 ports | Rigid | Dielectric sheet(2.33) |
| [24] | 0.91/1.57 | $0.48 \times 0.48 \times 0.015(1.7)$ * | NA/9.7% | LP/CP | O/U | $-10.96/8.26$ | 1 layer & 2 ports | Flexible | Felt (1.2,0.02) |
| [29] | 2.45/5.8 | $0.82 \times 0.82 \times 0.016(5.2)$ * | 4.6%/4.7% | CP/LP | U/O | 5.86/6.94 | 1 layer & 1 port | Flexible | Felt (1.2,0.02) |
| [32] | 2.38/5.8 | $\pi \times 0.28 \times 0.28 \times 0.024(2.8)$ * | 3.4%/4.3% | LP/LP | U/O | 4.16/3.66 | 1 layer & 1 port | Flexible | PDMS (2.8,0.02/ 0.04) |
| [33] | 2.45/5.0 | $\pi \times 0.2 \times 0.2 \times 0.033(2.0)$ * | 4.9%/15.7% | LP/LP | U/O | 5.5/5.3 | 2 layers & 1 port | Flexible | PDMS (2.7,0.013) |
| This work | 2.38/5.8 | $0.36 \times 0.36 \times 0.016(1)$ * | 2.5%/6.7% | CP/CP | O/U | $-2.8/6.8$ | 1 layer & 1 port | Flexible | Felt (1.2,0.02) |

Note: $\lambda_0$ = free space wavelength at the center frequency of the lower band; LP = linear polarization; CP = circular polarization; O = omnidirection; U = unidirection; * = normalized volume.

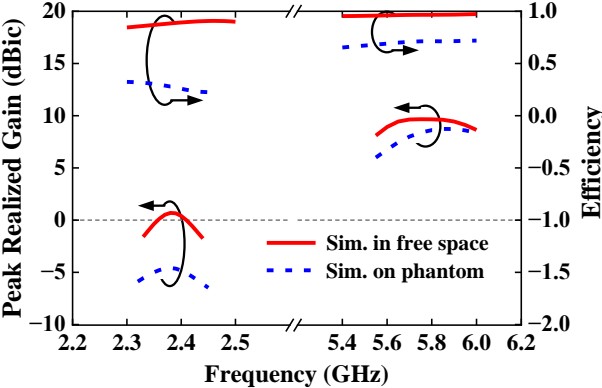

**Figure 22.** Simulated peak realized gain and radiation efficiency of the proposed antenna.

## 5. Conclusions

A CP textile antenna in two bands with a compact, mono-port, and single-layer configuration has been designed for concurrent on-/off-body wearable applications in the BAN. The comparison between previously reported wearable antennas and the proposed antenna is listed in Table 2; it is demonstrable that the proposed textile antenna acquires dual-band and dual-mode CP performance with the appropriate index, and, with the same felt material, the proposed antenna has the smallest sizes and is a flexible, comfortable material. By adopting the probes, stubs, and parasite elements, the omnidirectional CP radiation at 2.38 GHz and unidirectional CP radiation at 5.8 GHz are achieved. To clarify how the antenna works, its operating principle has been elaborated. The effects of structural deformation and the variation in the substrate material properties on antenna performance are thoroughly analyzed, revealing the robustness of the antenna. To validate the numerical results, the prototype was fabricated and measured in different scenarios, including the analysis of human body loading, bending, and humidity. As measured on the fresh pork, the $-10$ dB impedance bandwidths are averagely 2.5% (2.34–2.40 GHz) in the lower band and 6.7% (5.59–5.98 GHz) in the upper band, and the measured 3 dB AR bandwidths in the two bands are 2.5% (2.317–2.377 GHz) and 2.7% (5.658–5.814 GHz) with measured

peak realized gains of −2.8 and 6.8 dBic and radiation efficiencies of 25.2% and 67.7%, respectively. All the measured results of the proposed antenna are consistent with the simulated ones, verifying the validity of the numerical simulations. Additionally, through an SAR evaluation and communication capability measurement, it is assured that the proposed antenna provides acceptable transmission performance, making it a potential candidate for multifunctional operation in BANs.

**Author Contributions:** Conceptualization, X.L.; validation, Z.J.; investigation, Y.F. and H.Y. All authors have read and agreed to the published version of the manuscript.

**Funding:** This work was supported in part by the National Natural Science Foundation of China under Grant 61372008 and in part by the Guangdong Provincial Key Laboratory of Human Digital Twin under Grant 2022B1212010004.

**Informed Consent Statement:** Informed consent was obtained from all subjects involved in the study.

**Data Availability Statement:** The original contributions presented in this study are included in the article. Further inquiries can be directed to the corresponding author(s).

**Conflicts of Interest:** The authors declare no conflict of interest.

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
