# Peer review of "Dual Circularly Polarized Textile Antenna with Dual Bands and On-/Off-Body Communication Modes for Multifunctional Wearable Devices"

_electronics, doi:10.3390/electronics14091898_

Round 1
Reviewer 1 Report
Comments and Suggestions for Authors
The paper describe an iteresting proposal for a CP dual-band wearable antenna. A few points require minor corrections, to improve the usefulness of the paper. Moreover, there are many misuse of English terms (see "Comments on the Quality of English Language" for details). The list of required minor corrections follows.
All references must be enclosed in braces. This has been done only up to row 70.
Fig. 1: is too small for an antenna with so many details and dimensions. Please enlarge it, and enlarge also Fig. 9.
Fig. 2: at this point the details of the phantom used are useless. So, please, move Fig. 2 at the beginning of Sect. 3.1, and put the reference frame aside the figure, so that the axis can be clearly detected and all patterns shown in the paper can be clearly evaluated.
Fig. 6a: The input resistances of mode 2 and 3 are different. This reviewer imagines that this lead to a different excitation of those modes, with a negative effect on the CP. Please, check and comment.
row 307: the resonant frequency is only "almost proportional" to the permittivity, and only because the permittivity itself is quite small. Please, correct.
row 406: the increase in path loss from face-face link to face-side link is not due to a polarization mismatch, but to the antenna pattern. With reference to Fig. 10b, face-face link is at theta=0, where the maximum gain is located, while face-side link is at theta=90o , where the gain is about 15 dB below the maximum. This is the main reason for the increase of path loss. A polarization mismatch, which is clearly also present, lead to no more than 3-5 dB of loss.
Comments on the Quality of English Language
There are a too large number of misuse of English terms. And must be corrected, seeking the help of someone which known the technical English terms.
Many errors could be avoided reading with much more care the English textbook, instead of rely on Google Translate or likely SW.
The worst examples follow.
1) "aroused" instead of "excited". Google translate gived the same Chinese "word" for both terms. But, actually, in English, "aroused" means "raised, lift on", and has a sexual flavour. In any textbook the correct term is "excited".
2) "several" instead of "plural" in row 54
3) "remaining" instead of "left" in row 112
4) "ripple" (a technical term) instead of "out-of-roundness" in row 260
5) "phantom" instead of "manikin" in row 279
As a matter of fact, Google Translate is a good way to obtain a first translation of mothertongue terms, but then the result must be carefully checked on a good English Dictionary, such as the Oxford Dictionary.
Author Response
Please see the details in the attached word file. Thanks!

Reviewer 2 Report
Comments and Suggestions for Authors
The paper proposes a wearable antenna design on a felt substrate
- Most of the figures (ex. Figure 9, 17, 18, 19d, 20) with plots could be increased in size;
- Conclusions could have more summarized numerical data for the measurement results
- Commenting on "The substrate of the antenna uses the flexible felt with a dielectric constant of εr = 1.2 and a loss tangent of tan δ = 0.02. The radiation elements are composed of conductive nylon fabric with a thickness of 0.13 mm and a surface resistivity of less than 0.009 Ω/sq 35." - it would be a good idea to showcase the manufacturing of the proposed antenna. Is the conductive fabric glued to the felt substrate, is it sown to it is the flexible felt a commonly available substrate for factories and so on.
- In addition to Table 2, the mentioned similar works could include the type of dielectric used and the dielectric constant (it might be in a sentence commenting the table). The idea here is to have a rough comparison on the materials used in your case and other works.
All in all, I'd suggest accepting the paper with minor changes.
Author Response

(The authors gave the same response as above.)

Reviewer 3 Report
Comments and Suggestions for Authors
This research presents the design and experimental validation of a compact, dual-band, circularly polarized textile antenna with dual radiation modes for wearable body area networks. It introduces a novel, single-port, single-layer microstrip patch antenna that enables both omnidirectional on-body and unidirectional off-body communication, while maintaining robust performance under deformation, humidity, and proximity to human tissue.
Here are my comments:
The authors present current and field distributions to support the existence of TM₀₀, TM₁₀, TM_hp, and degenerate modes. There is no rigorous modal decomposition or eigenmode analysis to quantitatively confirm the excited modes and their orthogonality.
The antenna uses multiple elements, but coupling effects among them are not quantified.
Link-level performance metrics (e.g., BER under motion, fading margin, or energy efficiency in real use cases) are not analyzed.
No discussion of polarization degradation under dynamic motion or mismatch due to body proximity over time.
The lower band gain is notably low (–5.3 dBic).
This paper lacks reviewing recent studies in introdution, like Predicting flow status of a flexible rectifier using cognitive computing; and Funabot-Sleeve: A Wearable Device Employing McKibben Artificial Muscles for Haptic Sensation in the Forearm.
The integration method with garments is not elaborated. Is it thermally bonded? Sewn? What about mass manufacturability and consistency?
Textile conductivity stability over time (aging, corrosion) is unaddressed.
Author Response

(The authors gave the same response as above.)

Round 2
Reviewer 3 Report
Comments and Suggestions for Authors
The current version has addressed my concerns. I agree to publish it in It’s current form.